# Associations of Circulating Biomarkers with Disease Risks: A Two-Sample Mendelian Randomization Study

**DOI:** 10.3390/ijms25137376

**Published:** 2024-07-05

**Authors:** Abdulkadir Elmas, Kevin Spehar, Ron Do, Joseph M. Castellano, Kuan-Lin Huang

**Affiliations:** 1Department of Genetics and Genomic Sciences, Icahn School of Medicine at Mount Sinai, New York, NY 10029, USA; 2Nash Family Department of Neuroscience, Friedman Brain Institute, Icahn School of Medicine at Mount Sinai, New York, NY 10029, USA; 3Ronald M. Loeb Center for Alzheimer’s Disease, Icahn School of Medicine at Mount Sinai, New York, NY 10029, USA; 4Department of Neurology, Icahn School of Medicine at Mount Sinai, New York, NY 10029, USA; 5Black Family Stem Cell Institute, Icahn School of Medicine at Mount Sinai, New York, NY 10029, USA

**Keywords:** biomarkers, metabolome, proteome, mendelian randomization, human disease

## Abstract

Circulating biomarkers play a pivotal role in personalized medicine, offering potential for disease screening, prevention, and treatment. Despite established associations between numerous biomarkers and diseases, elucidating their causal relationships is challenging. Mendelian Randomization (MR) can address this issue by employing genetic instruments to discern causal links. Additionally, using multiple MR methods with overlapping results enhances the reliability of discovered relationships. Here, we report an MR study using multiple methods, including inverse variance weighted, simple mode, weighted mode, weighted median, and MR-Egger. We use the MR-base resource (v0.5.6) from Hemani et al. 2018 to evaluate causal relationships between 212 circulating biomarkers (curated from UK Biobank analyses by Neale lab and from Shin et al. 2014, Roederer et al. 2015, and Kettunen et al. 2016 and 99 complex diseases (curated from several consortia by MRC IEU and Biobank Japan). We report novel causal relationships found by four or more MR methods between glucose and bipolar disorder (Mean Effect Size estimate across methods: 0.39) and between cystatin C and bipolar disorder (Mean Effect Size: −0.31). Based on agreement in four or more methods, we also identify previously known links between urate with gout and creatine with chronic kidney disease, as well as biomarkers that may be causal of cardiovascular conditions: apolipoprotein B, cholesterol, LDL, lipoprotein A, and triglycerides in coronary heart disease, as well as lipoprotein A, LDL, cholesterol, and apolipoprotein B in myocardial infarction. This Mendelian Randomization study not only corroborates known causal relationships between circulating biomarkers and diseases but also uncovers two novel biomarkers associated with bipolar disorder that warrant further investigation. Our findings provide insight into understanding how biological processes reflecting circulating biomarkers and their associated effects may contribute to disease etiology, which can eventually help improve precision diagnostics and intervention.

## 1. Introduction

Molecular abnormalities detected in the blood that cause complex diseases represent an opportunity to identify biomarkers for both preventive and therapeutic interventions. These circulating biomolecules include proteins (enzymes and hormones), lipids, and metabolites that reflect physiological states of organ functions, immune response, and metabolism. Despite substantial observational evidence linking systemic biomarker levels with diverse health conditions, their causal relationships to complex diseases remain to be established, especially for diseases for which no reliable biomarkers exist. Previous studies that identify correlations often do not delineate cause and effect, thereby limiting the translational value of the findings. For example, while an increased incidence of impaired glucose metabolism has been demonstrated in patients with bipolar disorder across multiple studies, a causal relationship between the two has yet to be established [1,2,3].

The application of Mendelian Randomization (MR) represents an approach in addressing this critical gap. By utilizing genetic instruments as proxies for biomarker levels, MR, under specific assumptions, can control for confounding factors and reverse causation, offering insights into the causal effects of biomarkers on disease risks. Lipids have been particularly well studied in MR, where previous studies have demonstrated the causal relationships between LDL-c and coronary artery disease, and HDL-c and breast cancer [4,5,6,7]. Other studies have also identified likely causal relationships between homocysteine and stroke [8], metabolic syndrome [9], and tyrosine and Type 2 Diabetes [10]. However, the effects of many other biomarkers remain unclear, and existing MR studies have often been limited to specific biomarkers or disease categories.

The power of MR has been boosted by the availability of large genome-wide association studies (GWAS), published in more than 379 studies, including those conducted using the UK Biobank dataset that has recently released a resource of plasma biomarker data measured by nuclear magnetic resonance (NMR) in addition to other lab markers [11]. Herein, we conducted a comprehensive MR analysis encompassing a broad spectrum of 212 biomarkers—including 115 circulating biomolecules measured through NMR [12,13,14]—and 99 human diseases to unveil previously obscured relationships of the circulating biomarkers’ role in disease etiology that may be leveraged for personalized prevention.

## 2. Results

Based on the final 212 exposures (circulating biomarkers) and 99 outcomes (diseases) from the MRC IEU OpenGWAS database [15,16] (Section 4, Methods, STROBE-MR checklist), we analyzed exposure–outcome relationships using five different MR analysis methods: inverse variance weighted, MR-Egger, simple mode, weighted median, and weighted mode. To ensure our results were robust, we focused on findings that had a significant (Bonferroni *p*-value < 0.05) association by MR-Egger and matched in four or more methods (with same effect direction and raw *p*-value < 0.05) based on MR-Egger’s robustness to directional pleiotropy relevant to circulating biomarkers. We identified a total 21 significant biomarker exposure vs. disease outcome associations (Figure 1). A list of all significant relations, including IEU GWAS study IDs for each exposure/outcomes, and the summary statistics from the MR analyses are included in Appendix A.

We provide validation for our analysis by demonstrating agreement using four or more analysis methods that have been previously demonstrated in MR studies, such as urate with gout [17] and LDL [18], triglycerides [6], lipoprotein A [19], and apolipoprotein B [20] with coronary heart disease. We also found previously demonstrated relationships shown in MR studies between lipoprotein A [21] and LDL [22] with myocardial infarction. Additionally, we did not find any relationships that showed conflicting directionality to the previous literature. While many identified relationships are only maintained in a few methods, previously supported relationships show strong alignment in multiple MR analysis methods, demonstrating the robustness of associations found using multiple approaches.

For cardiometabolic traits, we demonstrated a causal relationship between total cholesterol and coronary heart disease, and this association has previously been shown in a meta-analysis and systematic review [23]. We also showed a causal relationship between total cholesterol and myocardial infarction as well as apolipoprotein B and myocardial infarction, both of which have been previously demonstrated by non-MR studies [24,25].

We also found a causal relationship between serum creatinine and chronic kidney disease (CKD). However, serum creatinine is an important biomarker for measuring kidney function, as it is used to clinically estimate glomerular filtration rate to help diagnose CKD [26,27]. While this connection could explain this result, there may be creatinine-specific effects that may cause or exacerbate CKD which remain to be characterized. We also identified a trend for alanine transferase with liver cell carcinoma that agreed for four or more methods when using raw *p*-value cut-offs of <0.05.

Of note, we discovered a strong, direct relationship between glucose and bipolar disorder and a strong, inverse relationship between cystatin C levels and bipolar disorder which, to the best of our knowledge, has never been directly reported before. Previous studies have demonstrated an association between impaired glucose metabolism and bipolar disorder in addition to increased prevalence of pre-diabetes and type 2 diabetes mellitus [3,28,29]. Table 1 lists the identified exposure–outcome relationships along with previous supporting literature and evidence type.

Figure 2 illustrates the top 16 significant (Bonferroni *p*-value < 0.05) biomarker exposure–disease outcome associations with agreement in effect directionality in four or more methods (effect size > |0.1|).

Several biomarkers are associated with an increased risk of disease. The average effect size of total cholesterol and coronary heart disease is 0.66, ranging from 0.90 ± 0.22 (MR-Egger) to 0.47 ± 0.13 (IVW). The association of total cholesterol and myocardial infarction has the average effect size of 0.61 with a range of (0.26, 1.13). For the apolipoprotein B and myocardial infarction, the average effect size is estimated as 0.59 with a range of (0.39, 0.90). For the glucose on bipolar disorder the average effect estimate is 0.39 with a range of (0.29, 0.55), while cystatin C has an inverse relationship on bipolar disorder with the average effect size of −0.31 ranging from −0.107 to −1 (Appendix A).

## 3. Discussion

We analyzed exposure-outcome relationships using five MR analysis methods to identify causal relationships between circulating biomarkers and diseases. In this study, significant results are defined as agreement between four or more analysis methods. Many of our demonstrated results replicated results from previous MR studies and other literature and none of our findings run contrary to previous literature. Additionally, our analysis also discovered two relationships of note: a direct relationship between glucose and bipolar disorder and an inverse relationship between cystatin C and bipolar disorder.

Bipolar disorder is a neuropsychiatric disorder with multifactorial causes (genetic, trauma, exposure to certain medications, etc.) Our findings demonstrate two potential circulating markers that may contribute to development of bipolar disorder, high glucose and low cystatin C. In our study, the effect size for glucose and bipolar disorder is 0.39 and for cystatin C and bipolar disorder is −0.31 which is comparable to the average effect size for triglycerides and coronary heart disease (a8) of 0.42 and greater than those of triglycerides and myocardial infarction (0.20), lipoprotein A and coronary heart disease (0.22), and lipoprotein A and myocardial infarction (0.21) (Appendix A). These results suggest that these relationships are likely to be of clinical significance and warrant further study. Although high cystatin C levels have been linked with major depressive disorder [32], no studies, to the best of our knowledge, have demonstrated an association between cystatin C and bipolar disorder. Levels of cystatin C, a natural inhibitor of cysteine proteases, are typically used clinically to assess kidney function. A common treatment for bipolar disorder is lithium, which has been shown to decrease renal function and can lead to elevated cystatin C levels [33], which may lend further insight to the mechanism behind lithium’s use as a treatment, however this remains to be confirmed.

A connection between impaired glucose metabolism and bipolar disorder has been well documented in the literature, with over half of patients diagnosed with bipolar disorder also having insulin resistance, impaired glucose tolerance, or type 2 diabetes [3]. Furthermore, it has been shown that modulating the PI3K/Akt insulin signaling pathway may be a mechanism of lithium for the treatment of bipolar disorder [34]. The PI3K/Akt pathway helps regulate glucose metabolism in the hippocampus, cerebellum, and olfactory bulb [35], which are vulnerable sites of gray matter loss in bipolar disorder [36]. It has been speculated that long-term disruption of this pathway can lead to mitochondrial dysfunction and energy dysregulation which may affect brain processes contributing to bipolar disorder [34]. Therefore, our identification of a causal relationship between high glucose levels and bipolar disorder further supports the current body of evidence, and our findings for cystatin C highlight a novel finding that warrants investigation in subsequent studies. Our findings suggest that future studies aiming to reduce the risk of bipolar disorder could explore the lifestyle and clinical interventions that reduce blood glucose and maintain kidney function.

Although our study focused on links that were aligned with four or more MR methods, other notable relationships were identified that were aligned with 2–3 MR methods (Figure 1). For example, C-reactive protein (CRP) and Alzheimer’s disease were causally linked in two methods (MR-Egger and IVW), which has been previously noted in previous non-MR studies [37].

A limitation of this study is that much of the exposure GWAS data comes from the UK biobank, a large-scale volunteer databank, which has a biased representation of Europeans and results may not generalize outside UK [38,39]. However, UK biobank provides the largest GWAS datasets of most biomarkers studied herein and adequate statistical power; this bias may be overcome as other biobanks with diverse populations continue to expand. MR studies have their own limitations, including but not limited to: potential confounding factors, limitations of estimating associations for binary outcomes, pleiotropic effects which should be mitigated by our use of MR-Egger, and population stratification [40,41,42].

Overall, by utilizing several MR methods as cross-validation, our study successfully reaffirmed several established connections in cardiovascular diseases, gout, and kidney diseases, while uncovering intriguing novel biomarkers that may be causal for bipolar disorder. These findings highlight the underappreciated roles that circulating biomarkers may play in disease mechanisms, which should open new avenues for targeted research and motivate development of precise diagnostic tools and therapeutic interventions.

## 4. Methods and Materials

### 4.1. Datasets

This study is a two-sample MR study based on publicly available GWAS summary statistics data from the MR-base resource (v0.5.6) [15]. The study design follows the Strengthening the Reporting of Observational Studies in Epidemiology using Mendelian Randomization (STROBE-MR) checklist, which is included in the Appendix A.

The MR-base platform, developed by the MRC Integrative Epidemiology Unit at the University of Bristol, serves both as a database and an analytical framework accessed through the TwoSampleMR R package (v0.5.7). MR-base offers access to multiple GWAS data, including the MRC Integrative Epidemiology Unit (IEU) OpenGWAS database consisting of 26,000 GWAS reported in at least 379 different studies, covering ~40 k individuals (in median) per study [15,16]. Based on MRbase, we utilized all accessible exposure traits grouped under the “Metabolites” category as well as other circulating biomarkers, and all available outcomes categorized as “Disease”. All diseases we used are from datasets with IDs beginning with “ieu” (GWAS summary datasets generated by many consortia curated by IEU) and “bbj” (Biobank Japan), and the biomarkers are from “ukb” (UK Biobank analyses by IEU or Neale lab covering 28 biomarkers) and “met” (human blood biomarkers, immune markers, and circulating biomarkers analyzed by Shin et al. 2014, covering 76 biomarkers, and Kettunen et al. 2016, covering 115 biomarkers) [12,14]. In instances where there are multiple summary datasets for the same trait (i.e., from different GWAS cohorts), we selected the top one from them based on the largest “sample size”, “year”, “number of SNPs”, or “number of cases and controls”, prioritizing the information available in this specified order. A list of all biomarkers used as exposures and information for their corresponding GWAS studies are available in Appendix A. Overall, we retained a total of 212 exposures (biomarkers) and 99 outcomes (diseases) for MR analyses.

Given that this is a two-sample MR study, we provide two major justifications: (1) Similarity of the genetic variant-exposure associations between the exposure and outcome samples: Genetic instruments for the exposure were primarily derived from GWAS of circulating biomarkers conducted using data from the UK Biobank in addition to Shin and Kettunen et al. [14], all of which predominantly consists of individuals of European ancestry. The outcome data were mainly sourced from case-control GWAS of European populations that likely share genetic architectures similar to those of the exposure study participants. (2) Number of individuals who overlap between the exposure and outcome studies: Given the separate sources of the genetic data for exposures and outcomes (various case-control studies), and the timing of data availability and publication, it is reasonable to assume minimal overlap in the individual participants across these studies. Particularly, all significant results were found based on exposure (biomarker) GWAS from UK Biobank, which has its first genetic data published in 2018. The significant outcome GWAS for bipolar disorder, cardiovascular disease (CVD), myocardial infarction (MI), gout, and chronic kidney disease all predated 2018. This temporal discrepancy between the data collection phases further supports the assumption of negligible overlap bias in our MR estimates.

### 4.2. MR Methods

The three core assumptions for Mendelian Randomization (MR) analysis are:

Relevance: The genetic instruments (e.g., single nucleotide polymorphisms, SNPs) used as instrumental variables (IVs) should be associated with the exposure (circulating biomarkers in this case). This assumption ensures that the IVs are strong enough to influence the exposure. Herein, 2685 SNPs were found associated with 198 circulating biomarker exposures subject to the IV assumptions that were imposed by the standard two-sample MR approach (association threshold, *p* < 5 × 10^−8^; LD clumping cutoff r^2^ > 0.001 within 10 Mb window) and were used for further analyses (Appendix A).

Independence: The genetic instruments should be independent of any confounding factors that may affect both the exposure and the outcome (disease). This assumption is justified by the random allocation of genetic variants. Further, each of the GWAS studies carefully control for covariates that may confound genetic associations. For example, in the UK Biobank biomarker GWAS conducted by the Neale lab, covariates include age, sex, age^2^, age × sex, age × sex^2^, and the first 20 PCs, making the final genetic instruments less prone to confounding. Additionally, the prior knowledge of demographic consistency can likely minimize the impact of unmeasured confounders due to similar demographic and genetic backgrounds.

Exclusion restriction: The genetic instruments should affect the outcome (disease) only through their effects on the exposure (circulating biomarkers) and not through any other direct or indirect pathways. This assumption ensures that the IVs influence the outcome solely via the exposure of interest. Critically, MR-Egger was employed as a primary MR method in this study to detect and adjust for pleiotropy, where genetic variants may influence the outcome via pathways other than through the exposure.

To enhance robustness, we inferred causality using multiple MR methods, specifically inverse variance weighted, simple mode, weighted mode, weighted median, and MR-Egger. The inverse variance-weighted (IVW) method conducts regression analysis between the SNP-exposure effect size and the SNP-outcome effect size, giving more importance to SNPs with the lowest standard error in SNP-outcome association. The IVW method relies on either all variants being valid instruments or the presence of balanced horizontal pleiotropy, where the combined horizontal pleiotropic effects of individual instruments cancel out. Additionally, it assumes that such pleiotropic effects are not dependent on instrument strength across all variants, a concept known as the Instrument Strength Independent of Direct Effects (InSIDE) assumption [43]. The simple mode method operates under the assumption that the most common value (i.e., mode) for horizontal pleiotropy is zero, known as the ZEro Modal Pleiotropy Assumption (ZEMPA), regardless of the specific type of horizontal pleiotropy [44]. Likewise, the weighted MBE method operates under the supposition that the most substantial weights in the k subsets originate from valid instruments, adhering to the ZEMPA assumption, irrespective of the particular form of horizontal pleiotropy. The weighted median method relies on the assumption that over half of the weight comes from reliable instruments, irrespective of the nature of horizontal pleiotropy [45]. MR-Egger permits all genetic variants to exhibit pleiotropic effects, but it necessitates that these effects are independent of the SNP-exposure associations, following the InSIDE principle [46]. Notably, the method can correct for the average horizontal pleiotropy observed across all variants considered in the study. Considering the recognized pleiotropic effects commonly seen in biomarkers and their relevance to health and disease [47], we used MR-Egger as the primary method for determining important exposure–outcome associations in our analyses and employed other sensitivity methods for validation.

For each method, the associations are reported with effect size and corresponding *p*-value statistics. The raw *p*-values are multi-testing corrected using the Bonferroni method. The confidence intervals (95%) are also calculated from standard errors of the test statistics. Overall, robustness of the results is ensured based on the use of MR-Egger as discovery method, agreement across at least 4 MR methods, and Bonferroni correction.

## Figures and Tables

**Figure 1 ijms-25-07376-f001:**
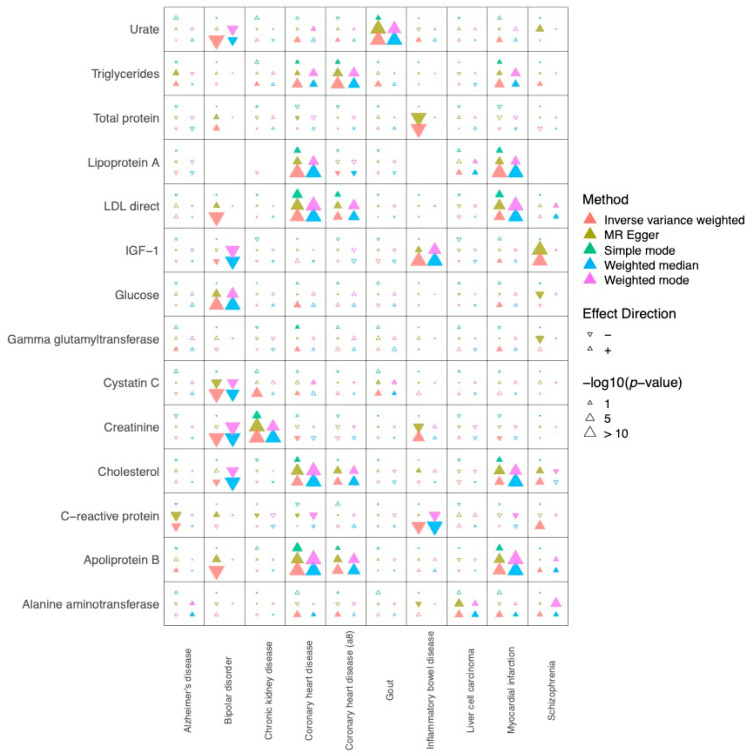
Heatmap summarizing the key findings of this study, including the most significant findings from the MR-Egger analysis. MR analyses were conducted using 5 methods: Inverse variance weighted, MR-Egger, Simple mode, Weighted Median, and Weighted Mode. Larger arrow sizes correspond to more significant results and the directionality of the arrow indicates a positively correlated relationship, i.e., higher biomarkers associated with increased risk (upward arrow) or inverse (downward arrow) relationship. We displayed all exposures and outcomes that demonstrated at least one significant (Bonferroni *p*-value < 0.05) relationship by at least one MR method.

**Figure 2 ijms-25-07376-f002:**
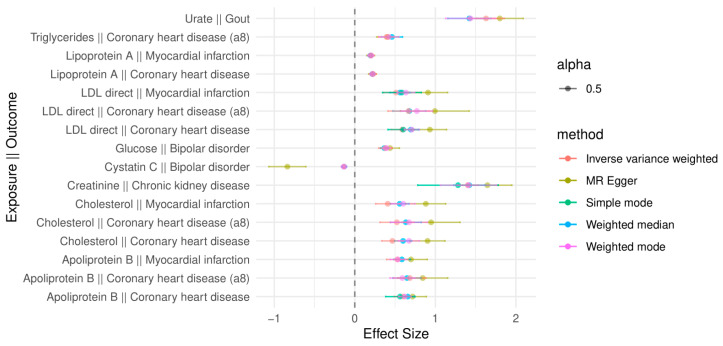
Forest plot of mendelian randomization analysis results assessing the causal effects of circulating biomarkers on diseases. The forest plot illustrates the estimated causal effects of the top 16 significantly associated (Bonferroni *p*-value < 0.05) biomarkers on the risk of various diseases based on Mendelian randomization (MR) analysis consistent across 4 or more methods. Each row represents a different MR association, with the biomarker as the exposure and the disease as the outcome. The points represent different effect size estimates from different MR methods and the 95% confidence intervals (based on standard errors of the effect size estimates) are displayed by the horizontal lines forming around the average effect sizes of different methods.

**Table 1 ijms-25-07376-t001:** Significant exposure–outcomes relationships consistently identified across 4 or more MR methods. Previous literature demonstrating these relationships and the type of evidence supporting each relationship are also detailed. In evidence type, retrospective, prospective, and cross-sectional indicates cohort studies that demonstrate associations of the biomarker with the disease outcome, whereas meta-analysis and systematic review represent syntheses of such studies that do not suggest causality.

Exposure	Outcome	Previous Evidence	Evidence Type
LDL	Coronary Heart Disease	[18]	MR
Triglycerides	Coronary Heart Disease	[6]	MR
Lipoprotein A	Coronary Heart Disease	[19]	MR
Apolipoprotein B	Coronary Heart Disease	[20]	MR
Cholesterol	Coronary Heart Disease	[23,30]	Meta-Analysis, Systematic Review, MR
Urate	Gout	[17]	MR
Lipoprotein A	Myocardial Infarction	[21]	MR
LDL	Myocardial Infarction	[22]	MR
Cholesterol	Myocardial Infarction	[24]	Retrospective
Apolipoprotein B	Myocardial Infarction	[25,31]	Prospective, MR
Serum Creatinine	Chronic Kidney Disease	[26,27]	MR, Meta-Analysis
Glucose	Bipolar Disorder	[28,29]	Cross Sectional, Meta-Analysis, Systematic Review
Cystatin C	Bipolar Disorder	N/A	N/A

## Data Availability

Data are contained within the article and Appendix A. GWAS summary statistics data used can be found on the MR-base resource (v0.5.6) through the ieu open GWAS project at the URL: https://gwas.mrcieu.ac.uk/ (accessed on 24 April 2024). The source code for the Mendelian randomization analyses is available at https://github.com/Huang-lab (accessed on 24 April 2024).

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
