# Peer review of "Associations of Circulating Biomarkers with Disease Risks: A Two-Sample Mendelian Randomization Study"

_ijms, 2024, doi:10.3390/ijms25137376_

Round 1

Reviewer 1 Report

Comments and Suggestions for Authors

I appreciate the opportunity to evaluate this paper. The paper fits my expertise well. The paper presents a Mendelian Randomization study (MR) of a series of 99 human ailments and 212 circulating biomarkers. The study is in general well written, with an easy to follow narrative. However, there are some important aspects I am concerned about.

First, the authors state in lines 47-49 that: “ Previous studies that identify correlations often do not delineate case and effect, thereby limiting the translations value of the findings.” I cannot agree more with the statements. This is a huge issue of the -omics revolution that started in the last 25 years. I am guilty of that; I have done studies that report findings but fall short of direct translational application. However, the authors are just doing that here. The most that can be possible offered from this study is the same level of correlational inference.

Second, how are the findings in this study novel and are not just reporting what is already known? We can even forget the causality for a minute and just focus on the mere associations, I am not sure, but these don’t seem novel.

Third, and following my first two arguments. The authors note in their limitations (line 275-280) that there is an important bias towards UKBiobank data. As they noted, this is because it is the single most important repository. Therefore, how is this study not just some sort of an internal validation of UKBiobank findings?

Two technical questions:

I am not sure I understand the rationale for selecting those 99 ailments and 212 circulating biomarkers.

The study looks a like a multiple testing nightmare with a large number of simultaneous tests on a long list of ailments. I may have missed any experiment-wise adjustments, and I may have misinterpreted the statement on line 117-118 on significance being a mere <0.05.

In summary, the biggest issues with this paper are conceptual on two routes. What is this study solving and what did the study find that is novel? I am not sure if either of these paths are being advanced. I would like to offer the opportunity to the authors to respond to my critique, but I would require a significant revision in the narrative of the paper to motivate me to recommend this paper for publication.

Author Response

Authors: We thank the reviewers for their thoughtful critiques and the opportunity to address the concerns raised. Our study aims to go beyond mere association by leveraging multiple Mendelian Randomization (MR) methods to infer causal relationships between circulating biomarkers and complex diseases. Below, we address each point in detail.

Reviewer 1:

I appreciate the opportunity to evaluate this paper. The paper fits my expertise well. The paper presents a Mendelian Randomization study (MR) of a series of 99 human ailments and 212 circulating biomarkers. The study is in general well written, with an easy to follow narrative. However, there are some important aspects I am concerned about.

First, the authors state in lines 47-49 that: “ Previous studies that identify correlations often do not delineate case and effect, thereby limiting the translations value of the findings.” I cannot agree more with the statements. This is a huge issue of the -omics revolution that started in the last 25 years. I am guilty of that; I have done studies that report findings but fall short of direct translational application. However, the authors are just doing that here. The most that can be possible offered from this study is the same level of correlational inference.

Authors: We thank the reviewer for their thoughtful critique and the opportunity to address the concerns raised. Our study aims to go beyond mere association by leveraging multiple Mendelian Randomization (MR) methods to infer causal relationships between circulating biomarkers and complex diseases. By utilizing genetic variation as instrumental variable, MR can demonstrate causation and therefore provides more than a baseline correlational inference.

Second, how are the findings in this study novel and are not just reporting what is already known? We can even forget the causality for a minute and just focus on the mere associations, I am not sure, but these don’t seem novel.

Authors: We acknowledge the importance of distinguishing our findings from known associations. Our study does confirm several established relationships, which serves as a validation of our methods. However, we also report novel associations. The findings of a causal relationship between glucose and bipolar disorder as well as cystatin C and bipolar disorder are novel findings by MR. Additionally, the findings between cystatin C and bipolar disorder have not been identified previously in any form.

Third, and following my first two arguments. The authors note in their limitations (line 275-280) that there is an important bias towards UKBiobank data. As they noted, this is because it is the single most important repository. Therefore, how is this study not just some sort of an internal validation of UKBiobank findings?

Authors: We thank reviewer for this comment. To the best of our knowledge, these are not associations directly found in the UKBiobank. This is an extended analysis of the resulting biomarker GWAS data and other disease GWAS that provides meaningful insights into potential causal relationships between blood-based biomarkers and disease. However, there may be biases in the findings as much of the data is UK-specific and may be biased by the background of individuals in the database.

Two technical questions:

I am not sure I understand the rationale for selecting those 99 ailments and 212 circulating biomarkers.

Authors: These were all of the diseases and biomarkers in the database with available GWAS data for MR analyses. This (limitation of the data resource) is stated explicitly in the Methods section, lines 170-172.

The study looks a like a multiple testing nightmare with a large number of simultaneous tests on a long list of ailments. I may have missed any experiment-wise adjustments, and I may have misinterpreted the statement on line 117-118 on significance being a mere <0.05.

Authors: We acknowledge the multiple testing issue inherent in our study. To address this, we employed a very stringent significance threshold and performed Bonferroni correction to account for the multiple comparisons. Specifically, we adjusted our significance level to α=0.05/(number of tests), ensuring that our findings are robust against type I errors. This adjustment is explicitly mentioned in our methods section and now clarified on page 2 in lines 75-78 as well as on page 7 in lines 235-239: “The raw p-values are multi-testing corrected using the Bonferroni method. The confidence intervals (95%) are also calculated. Overall, robustness of the results is ensured based on the use of MR-Egger as discovery method, agreement across at least 4 MR methods, and Bonferroni correction.

In summary, the biggest issues with this paper are conceptual on two routes. What is this study solving and what did the study find that is novel? I am not sure if either of these paths are being advanced. I would like to offer the opportunity to the authors to respond to my critique, but I would require a significant revision in the narrative of the paper to motivate me to recommend this paper for publication.

Authors:

What is this study solving: This study addresses the challenge of establishing causal relationships between circulating biomarkers and complex diseases. While many studies have identified associations between biomarkers and diseases, they do not entail causality, i.e., changed biomarkers can be a consequence of the disease. By using Mendelian Randomization (MR) with multiple methods, we could advance the understanding of how biomarkers contribute to disease mechanisms.

What is novel: This study found causal relationships between glucose and bipolar disorder as well as cystatin C and bipolar disorder which are novel findings by MR. Additionally, the findings between cystatin C and bipolar disorder have not been identified previously in any form. We have also updated the Abstract and minorly main texts to clarify this narrative as the reviewer suggested.

Reviewer 2 Report

Comments and Suggestions for Authors

The manuscript provides valuable insights into the causal associations between circulating biomarkers and complex diseases utilizing multiple MR methodologies. However, several areas of the study may benefit from further clarification and adjustments:

1. The link between glucose and bipolar disease through the PI3K-Akt pathway is intriguing but not sufficiently detailed. Expanding on how this pathway may mediate the observed effects could strengthen the causal inference.

2. The current formatting of the references may confuse readers, particularly concerning the numbering and citation style. 

3. The manuscript employs various p-value thresholds across different analytical methods. An explanation of the rationale behind these choices and their impact on the study outcomes would aid in understanding the robustness and reliability of the findings.

4. More detailed discussion on the magnitude of effect sizes reported and their biological or clinical significance could provide deeper insights into the relevance of the biomarker-disease associations identified.

5. Please include specific versions of the statistical packages and software used in the analyses ensures reproducibility and credibility of the results.

6. The Institutional Review Board (IRB) statement appears to be still a generic IJMS template. 

Author Response

Authors: We thank the reviewers for their thoughtful critiques and the opportunity to address the concerns raised. Below, we address each point in detail.

Reviewer 2:

The manuscript provides valuable insights into the causal associations between circulating biomarkers and complex diseases utilizing multiple MR methodologies. However, several areas of the study may benefit from further clarification and adjustments:

  1. The link between glucose and bipolar disease through the PI3K-Akt pathway is intriguing but not sufficiently detailed. Expanding on how this pathway may mediate the observed effects could strengthen the causal inference.

Authors: This portion of the discussion has been expanded in the text on pages 7 and 8, by stating “The PI3K/Akt pathway helps regulate glucose metabolism in the hippocampus, cerebellum, and olfactory bulb40 which are vulnerable sites of gray matter loss in bipolar disorder.41 It has been speculated that long-term disruption of this pathway can lead to mitochondrial dysfunction and energy dysregulation which may affect brain processes contributing to bipolar disorder.39

  1. The current formatting of the references may confuse readers, particularly concerning the numbering and citation style. 

Authors: Citations have been fixed.

  1. The manuscript employs various p-value thresholds across different analytical methods. An explanation of the rationale behind these choices and their impact on the study outcomes would aid in understanding the robustness and reliability of the findings.

Authors: We applied the Bonferroni correction to p-values and used a standard threshold of 0.05 to report significant findings (which is clarified on page 2 in lines 75-78 as well as on page 7 in lines 235-239.). The raw p-values are only reported as supporting evidence of consistency between multiple sensitivity methods of MR (clarified on page 2, lines 75-77). We further clarified the lines 125-127 and improved writing in the corrected statement intending to highlight the “level” of significance reached in the top-20 significant results (instead of some “threshold” of significance) as “Figure 2 illustrates the top-20 significant (p-value < 0.05) biomarker exposure-disease outcome associations with agreement in effect directionality in 4 or more methods (effect size > |0.1|)”.

  1. More detailed discussion on the magnitude of effect sizes reported and their biological or clinical significance could provide deeper insights into the relevance of the biomarker-disease associations identified.

Authors: A discussion and interpretation of the effect sizes and their relevance was added to the discussion section on page 7 lines 251-255: “In our study, the effect size for glucose and bipolar disorder is 0.39 and for cystatin C and bipolar disorder is -0.31 which is comparable to the average effect size for triglycerides and coronary heart disease (a8) of 0.42 and greater than those of triglycerides and myocardial infarction (0.20), lipoprotein A and coronary heart disease (0.22), and lipoprotein A and myocardial infarction (0.21).”.

  1. Please include specific versions of the statistical packages and software used in the analyses ensures reproducibility and credibility of the results.

Authors: We used the MR-base resource (v0.5.6) and the TwoSampleMR R package (v0.5.7). The source codes for the MR analyses performed in this study are available at www.github.com/Huang-lab/MR.  The Methods section has been updated to reflect the version of the R package used.

  1. The Institutional Review Board (IRB) statement appears to be still a generic IJMS template. 

Authors: We have now updated the statement,

Institutional Review Board Statement: Not applicable. Our analyses were conducted using publicly available summary statistics data from GWAS, not individual-level data; therefore, this is a non-human subject research that requires no additional ethical approval.

Round 2

Reviewer 1 Report

Comments and Suggestions for Authors

I appreciate the opportunity to review a minimally revised version of the paper. Based on the extent of your revision and the fact that the journal immediately sent it back to me for a second review, I can assume my arguments and concerns do not have much weight on the overall decision that will be taken.

I continue to believe that this paper provides little advance to the field and commits the same offense it complains regarding only offering biomarker lists. I disagree with the associations found between glucose and bipolar disorder and cystatin C and bipolar disorder being enough to suggest causality. But considering that my original concerns not bearing too much weight on the revisions I see here, I suppose that might not be a priority for this paper.

However, I must admit I was wrong on the multiple testing part. It is indeed mentioned in the methods, but not in the figures or elsewhere. I am not sure if the 95% CI that are barely visible in the fuzzy figures provided are the raw or corrected. I would suggest that you label that these adjustments are considered in the figures.

Outside of these remarks I don’t see a major concern with the publication of this paper.

Good luck with your research.

Author Response

Comment:

I appreciate the opportunity to review a minimally revised version of the paper. Based on the extent of your revision and the fact that the journal immediately sent it back to me for a second review, I can assume my arguments and concerns do not have much weight on the overall decision that will be taken.

I continue to believe that this paper provides little advance to the field and commits the same offense it complains regarding only offering biomarker lists. I disagree with the associations found between glucose and bipolar disorder and cystatin C and bipolar disorder being enough to suggest causality. But considering that my original concerns not bearing too much weight on the revisions I see here, I suppose that might not be a priority for this paper.

However, I must admit I was wrong on the multiple testing part. It is indeed mentioned in the methods, but not in the figures or elsewhere. I am not sure if the 95% CI that are barely visible in the fuzzy figures provided are the raw or corrected. I would suggest that you label that these adjustments are considered in the figures.

Outside of these remarks I don’t see a major concern with the publication of this paper.

Good luck with your research.

Response:
We understand the reviewer's concerns regarding the significance of our findings. Our intention was to provide a comprehensive biomarker analysis that highlights potential associations through MR, which is an established method to establish causality. We particularly used 4 methods to ensure the robustness of our findings. We agree further studies need to be conducted to address the limitations of our study, which is extensively discussed in the Discussion. 

We also thank the reviewer for the mention of multiple testing in our methods. To further enhance clarity, we have revised Figure 2 to include only associations passing Bonferroni correction for stringency. As in the previous draft, only those associations passing Bonferroni correction are discussed in the text. We have also ensured that the 95% confidence intervals in the figures are clearly labeled. Additionally, we will provide high-quality, separately attached figures for better clarity in publication. The previous versions in word doc may be reduced in size.